# Gold Nanoparticle DNA Damage by Photon Beam in a Magnetic Field: A Monte Carlo Study

**DOI:** 10.3390/nano11071751

**Published:** 2021-07-03

**Authors:** Mehwish Jabeen, James C. L. Chow

**Affiliations:** 1Department of Physics, Ryerson University, Toronto, ON M5B 2K3, Canada; mehwish.jabeen@ryerson.ca; 2Department of Radiation Oncology, University of Toronto and Radiation Medicine Program, Princess Margaret Cancer Centre, Toronto, ON M5G 1Z5, Canada

**Keywords:** gold nanoparticle, nanoparticle-enhanced radiotherapy, MR-guided radiotherapy, DNA damage, Monte Carlo simulation, dose enhancement, magnetic field

## Abstract

Ever since the emergence of magnetic resonance (MR)-guided radiotherapy, it is important to investigate the impact of the magnetic field on the dose enhancement in deoxyribonucleic acid (DNA), when gold nanoparticles are used as radiosensitizers during radiotherapy. Gold nanoparticle-enhanced radiotherapy is known to enhance the dose deposition in the DNA, resulting in a double-strand break. In this study, the effects of the magnetic field on the dose enhancement factor (DER) for varying gold nanoparticle sizes, photon beam energies and magnetic field strengths and orientations were investigated using Geant4-DNA Monte Carlo simulations. Using a Monte Carlo model including a single gold nanoparticle with a photon beam source and DNA molecule on the left and right, it is demonstrated that as the gold nanoparticle size increased, the DER increased. However, as the photon beam energy decreased, an increase in the DER was detected. When a magnetic field was added to the simulation model, the DER was found to increase by 2.5–5% as different field strengths (0–2 T) and orientations (x-, y- and z-axis) were used for a 100 nm gold nanoparticle using a 50 keV photon beam. The DNA damage reflected by the DER increased slightly with the presence of the magnetic field. However, variations in the magnetic field strength and orientation did not change the DER significantly.

## 1. Introduction

In radiotherapy, the aim is to acquire a conformal dose at the tumor or target as high as possible while, at the same time, sparing the surrounding normal tissues at the minimum dose. One way to achieve this goal is to add a heavy-atom radiosensitizer such as gold nanoparticles to the tumor to increase its compositional atomic number [1,2,3]. This increase in radiosensitivity is due to a combination of the physical dose enhancement and additional chemical and biological effects associated with the nanoparticle [4]. Gold nanoparticles can be transported to living cells through a liposome-based system. This makes the treatment delivery of gold nanoparticle-enhanced radiotherapy possible [5]. There are two advantages of adding gold nanoparticles to the tumor. First, the increase in the compositional atomic number of the tumor increases the radiation dose absorption through the enhancement of the photoelectric effect. This dose enhancement is particularly significant when photon beam energy in the kilovoltage (kV) range is used, where the photoelectric effect is dominant [6,7,8]. Moreover, the dose enhancement can increase cancer cell killing. Second, due to the increase in the deviation of beam absorption between the target (with gold nanoparticle addition) and its surrounding tissue (without gold nanoparticle addition), contrast enhancement can be achieved in medical imaging modalities such as computed tomography (CT) using a kV photon beam [9,10,11]. The increase in target contrast will make it easier for the radiation oncologist to identify the tumor and contour it more accurately in radiation treatment planning. Therefore, gold nanoparticle-enhanced radiotherapy has become popular, resulting in many studies on the basic science and clinical application [12,13,14,15]. Many preclinical works have been carried out, and clinical trials are being conducted, building a potential roadmap to clinical implementation [16].

In radiobiology, it is well known that cancer cell killing or control is caused by the energy deposition from the secondary electrons at the deoxyribonucleic acid (DNA). These secondary electrons generated by the ionizing radiation in the tumor medium (water equivalent) would travel to the strands of DNA and damage the molecule, for example, causing a single- and double-strand break [17,18]. The addition of gold nanoparticles to the tumor can enhance the energy deposition because extra secondary electrons are produced by the irradiated nanoparticles. Therefore, more lethal double-strand breaks would be produced in the DNA [19]. Reproduction of the cancer cell is therefore terminated because the DNA has been damaged by the radiation and irradiated gold nanoparticles.

Recently, with advances in magnetic resonance (MR)-guided radiotherapy [20,21,22], MR images can be acquired during radiation dose delivery. This allows radiation staff to examine the patient’s tumor when it is irradiated by photon beams. Moreover, MR imaging can provide an excellent contrast of soft tissue compared to the routine CT imaging. This results in better tumor contouring and targeting [23]. To date, MR-guided radiotherapy has improved soft tissue visualization, management of the intrafraction and interfraction organ motion and online adaptive radiotherapy [24]. However, there is a concern over the dose distribution, affected by the magnetic field from the MR system, in the patient [25]. The absorbed dose contributing to cancer cell killing is determined by the energy deposition in the DNA, due to the secondary electrons generated from the interaction between the radiation beam and tumor medium. Since an electron is a charged particle and its traveling path is affected by the magnetic field, the electron distribution in the tumor can be affected by the presence of a magnetic field, leading to a change in final dose distribution. This may, in turn, affect the treatment outcome of MR-guided gold nanoparticle-enhanced radiotherapy. Although there are macroscopic studies concerning the variation in dose distribution due to the magnetic field in MR-guided radiotherapy [26,27,28], there is a lack of study on the nanodosimetry regarding the DNA damage, not to say with gold nanoparticle addition.

In this study, we investigated this problem by focusing on the nanodosimetry of a gold nanoparticle and DNA. Using Monte Carlo simulation, we examined the dose enhancement of DNA in the presence of a gold nanoparticle when a magnetic field is or is not added to the beam irradiation. Through determining the dose enhancement factor (DER) at the DNA with different nanoparticle sizes, photon beam energies and magnetic field strengths and orientations, we can find out the relationship between the DNA damage and the presence of the magnetic field when an irradiated gold nanoparticle interacts with a DNA.

## 2. Materials and Methods

### 2.1. Monte Carlo Simulation

Monte Carlo simulation was used to investigate the influence of the magnetic field on dose enhancement in the DNA when a gold nanoparticle was irradiated by a photon beam. Monte Carlo simulation is a widely used mathematical method in medical physics to model radiation techniques, assess the dose distribution and analyze radiation effects in a certain environment under different experimental conditions [29,30]. In this study, Geant4 software developed by CREN was used to conduct Monte Carlo simulation [31]. The source code, Geant4-DNA, is an extension of the Geant4 Monte Carlo toolkit used to simulate the irradiation of gold nanoparticles and DNA with a photon beam [32]. The code can construct the environment of a gold nanoparticle near a DNA irradiated by a photon beam at a distance from the nanoparticle in the presence of a magnetic field. Geant4-DNA provides a virtual machine containing CentOS Linux, and the latest version of Geant4 (version 10.7), analysis tools, visualization tools and other utilities were used in the simulation. VMware Workstation 16 player was used for running the virtual machine. This machine consists of a Linux distribution of CentOS 8 64-bit. The virtual machine has pre-installed codes and all the software required to run Geant4. The virtual machine was installed from the CREN website (https://geant4.cenbg.in2p3.fr/ accessed date: 1 January 2021).

The default DNA physics list class, “G4EmDNAPhysics_option2”, was implemented in this study, which is recommended for cellular-scale simulations. It includes several physics models that cover physical interactions needed for particle transport in water medium [33]. Different physics models have to be defined in gold nanoparticles for different particles such as photons and electrons. Since the transportation of particles in Geant4-DNA is only valid in water medium, a macroscopic physics list, for example, “G4LivemorePhysics”, is defined for physical interactions of photons with gold medium. In this study, the environmental model of the cell was assumed to be water equivalent.

### 2.2. Simulation Model and Geomtry

A DNA model according to Henthorn et al. [34], alongside a gold nanoparticle, was defined inside a spherical water phantom with a radius of 0.5 µm. The simulation variables were similar to Chun et al. [35]. In this model, the backbones and bases were constructed as tiny spheres with a radius of 0.24 nm and 0.208 nm, respectively. Figure 1 shows the simulation setup for the study. The radiation source was defined as a circular plane source with a radius of twice the radius of the gold nanoparticle. The three photon beam energies considered for this simulation were 50, 100 and 150 keV. The primary photons emitted from the left side of the gold nanoparticle reached the DNA molecule. The most important cause of energy deposition in the DNA is the secondary electrons emitted from the gold nanoparticle. In this study, different nanoparticle diameters (30, 50 and 100 nm) with a nanoparticle-to-DNA distance of 30 nm were used. Photon beam energies of 50, 100 and 150 keV were used with a uniform magnetic field (0, 1 and 2 T) defined along each axis, in order to examine the effect of the magnetic field on the dose enhancement. In Figure 1, three orientations were considered for the magnetic field (B_x_, B_y_ and B_z_), separately. Since the primary photons were emitted along the z-axis, the magnetic field orientation parallel to the z-axis was also parallel to the trajectory of the photon, and the magnetic field orientation parallel to the x-axis and y-axis was perpendicular to the trajectory of the photon.

Figure 2 shows electron tracks in the simulation model. Energy deposition happened when secondary electrons (yellow dots) were generated along the electron tracks (red). If energy deposition occurred in the DNA (i.e., right-hand side of Figure 2), ionization of the strand of DNA may happen, leading to DNA damage [9,19,36]. The number of histories for the photons interacting with the gold nanoparticle was equal to 300 million in this study, and more photons (~2 billion) were required to achieve a similar uncertainty (2–5% standard deviation) for the simulation model without the gold nanoparticle.

### 2.3. Dose Enhacnement Ratio

The enhancement of energy deposition in the presence of gold nanoparticles, resulting in DNA damage, can be expressed as the DER [37]:$$\mathrm{Dose}\;\mathrm{Enhancement}\;\mathrm{Ratio}\;\left(\mathrm{DER}\right)=\frac{\mathrm{Dose}\;\mathrm{in}\;\mathrm{the}\;\mathrm{DNA}\;\mathrm{with}\;\mathrm{gold}\;\mathrm{nanoparticle}\;\mathrm{addition}}{\mathrm{Dose}\;\mathrm{in}\;\mathrm{the}\;\mathrm{DNA}\;\mathrm{without}\;\mathrm{gold}\;\mathrm{nanoparticle}\;\mathrm{addition}}$$

When no gold nanoparticle was added to the simulation model, the material of the particle was changed from gold to water. This mimicked the environment of a homogeneous tumor with water equivalent. The DER is therefore equal to one. A DER greater than one shows a dose enhancement due to the gold nanoparticle addition.

## 3. Results and Discussion

The relationships between the DER and different simulation variables, namely, nanoparticle size and magnetic field strength and orientation for photon beam energies of 50, 100 and 150 keV, are shown in Figure 3a–c, respectively. The gold nanoparticle diameters were equal to 30, 50 and 100 nm, and the magnetic field strengths were equal to 0, 1 and 2 T, with orientations parallel to the x-, y- and z-axis, as shown in Figure 1. The distance between the nanoparticle and DNA was equal to 30 nm.

### 3.1. Dependence of DER on Nanoparticle Size and Beam Energy

When there was no magnetic field present in the simulation model (i.e., magnetic field strength = 0), the DER was found to increase with the gold nanoparticle size. The rates of increase in the DER were 3.4%, 4.5% and 2.9%/nm for photon beam energies equal to 50, 100 and 150 keV, respectively (Figure 3a–c). The maximum DER was found to be 7.16 for the gold nanoparticle with a diameter equal to 100 nm using the 50 keV photon beam, while the minimum DER was 3.59 for the nanoparticle with a 30 nm diameter using the 150 keV beam. The reason for an increase in the DER with an increase in the nanoparticle size is that the larger particle contains more gold atoms to interact with photons in order to produce secondary electrons, resulting in more energy deposition in the DNA [6]. However, when the nanoparticle size becomes larger, the self-absorption of electrons in the nanoparticle also becomes more significant. This self-absorption effect would decrease the DER when the nanoparticle size increases [38].

For the same nanoparticle size with various photon beam energies, it was found that the DER decreased with a beam energy increase. For the nanoparticle with a 50 nm diameter, as shown in Figure 3a–c, the DER was found to decrease from 6.09 to 4.55 when the photon beam energy increased from 50 to 150 keV. This can be explained by the enhancement of the photoelectric effect as the cross-section or attenuation coefficient of the photoelectric interaction decreased with an increase in the photon energy [7,8,9]. The trends of variation in the DER on the nanoparticle size and photon beam energy agreed with results of our previous work using an older DNA model in the simulation [35].

### 3.2. Dependence of DER on Magnetic Field Strength and Orientation

For the nanoparticle with a 100 nm diameter using the 50 keV photon beam, as shown in Figure 3a–c, the DER was found to be 7.16, 7.35 and 7.36 when the magnetic field strength was equal to 0, 1 and 2 T along the z-axis. These were increases of about 2.5% and 2.7% in the DER when a magnetic field of 1 and 2 T was added to the simulation model along the z-axis (Figure 1). Similar increases in the DER could be found in the nanoparticles with diameters equal to 30 and 50 nm. It is found that the presence of the magnetic field along the central beam axis would increase the energy deposition in the DNA slightly. However, the increase in the magnetic field strength did not lead to a significant increase in energy deposition in the DNA.

When considering the nanoparticle with a 100 nm diameter using the 50 keV photon beam, with the magnetic field perpendicular to the central beam axis, but parallel to the DNA (i.e., x-axis in Figure 1), the DER was found to be 7.16, 7.36 and 7.46 when the magnetic field strength was equal to 0, 1 and 2 T (Figure 3a–c). These were increases of about 2.7% and 4.0% when a magnetic field of 1 and 2 T was added to the simulation model along the x-axis. Similar increases in the DER were found for nanoparticles of 30 and 50 nm diameter. However, the increase in the DER was in the range of 0.50–1.8%, which was smaller than the nanoparticle with a 100 nm diameter. It is seen that the presence of a magnetic field perpendicular to the central beam axis would increase the energy deposition in the DNA more than along the central beam axis, and the increase was more significant for the larger nanoparticle.

With the same photon beam energy of 50 keV, with the magnetic field along the y-axis (Figure 1), the orientation of the field was perpendicular to the central beam axis and the DNA. This beam and magnetic field geometry was different from placing the magnetic field along the x-axis because the DNA was parallel to the x-axis of the photon beam. For the nanoparticle with a diameter of 100 nm, the DER was found to increase by 2.8% and 4.9% when the magnetic field strength increased by 1 and 2 T, respectively (Figure 3a–c). This increase in energy deposition in the DNA was very similar to the magnetic field orienting in the x-axis. It is seen that the orientation of DNA did not affect its energy deposition significantly regarding the magnetic field orientation.

This work focused on the nanodosimetric change in the DNA interacting with an irradiated gold nanoparticle in the presence of a magnetic field. Based on the results in this study, it is worthwhile to further investigate the dependence of the DER on multiple nanoparticles with different distribution patterns, sizes and shapes. Moreover, macroscopic Monte Carlo simulation [39] can be carried out to investigate the dose enhancement of a tumor in a patient treated with magnetic resonance-guided radiotherapy. This multi-scale study can help us to understand, in detail, the impact of the magnetic field on cancer control from the nanometer to centimeter scales.

## 4. Conclusions

The DER determined, based on the energy deposition in the DNA, for varying gold nanoparticle diameters (30, 50 and 100 nm), photon beam energies (50, 100 and 150 keV) and magnetic fields (0 T, 1 T and 2 T) in the x, y and z orientations was investigated using Monte Carlo simulation. In general, the DER with no magnetic field present increased as the gold nanoparticle size increased. Moreover, the DER decreased as the photon beam energy increased, since the secondary electrons generated at high energies are less than the electrons generated at lower energies. In the presence of a magnetic field, the DER increased by about 2.5–5% for various field strengths (1 and 2 T) and orientations (x-, y- and z-axis) for the largest nanoparticle (diameter equal to 100 nm) using the lowest photon beam energy (50 keV) in this study. It was found that the increase in the DER was even smaller for smaller gold nanoparticles using a higher photon beam energy.

The results in this work provide important information concerning the variation in energy deposition in DNA when a magnetic field is present in an irradiated gold nanoparticle. Moreover, this single-nanoparticle model can act as a base for the construction of a more complicated multi-nanoparticle model focusing on a more realistic cellular environment for clinical practice.

## Figures and Tables

**Figure 1 nanomaterials-11-01751-f001:**
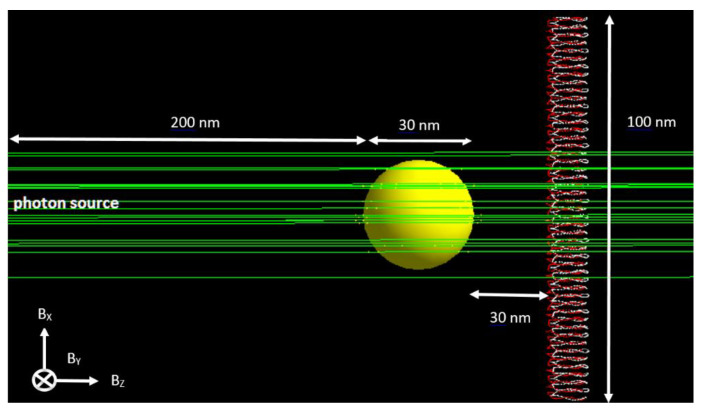
The Monte Carlo model geometry simulated in Geant4-DNA (not to scale). The gold nanoparticle was placed between the photon beam (green) and the DNA molecule. Nanoparticle diameters of 30, 50 and 100 nm were used in the simulation.

**Figure 2 nanomaterials-11-01751-f002:**
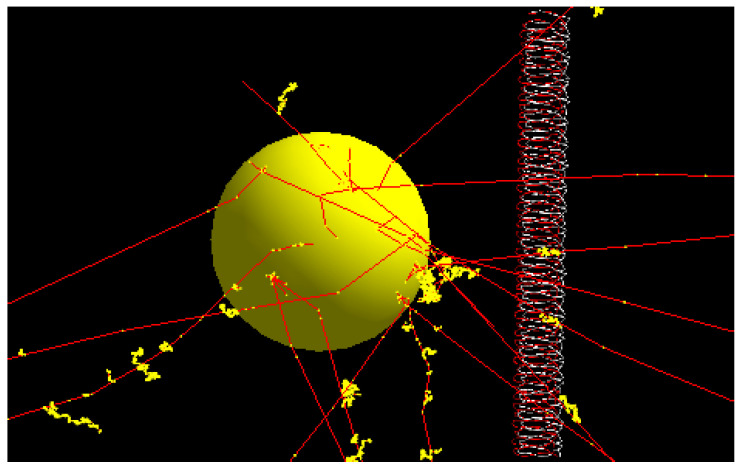
Schematic diagram showing the gold nanoparticle irradiated by the photon beam with electron tracks (red) generated from the nanoparticle. Energy deposition happens when the secondary electrons (yellow dots) are generated along the electron paths. When energy deposition occurs in the DNA, lethal DNA damage such as a double-strand break may be produced, leading to cancer cell killing.

**Figure 3 nanomaterials-11-01751-f003:**
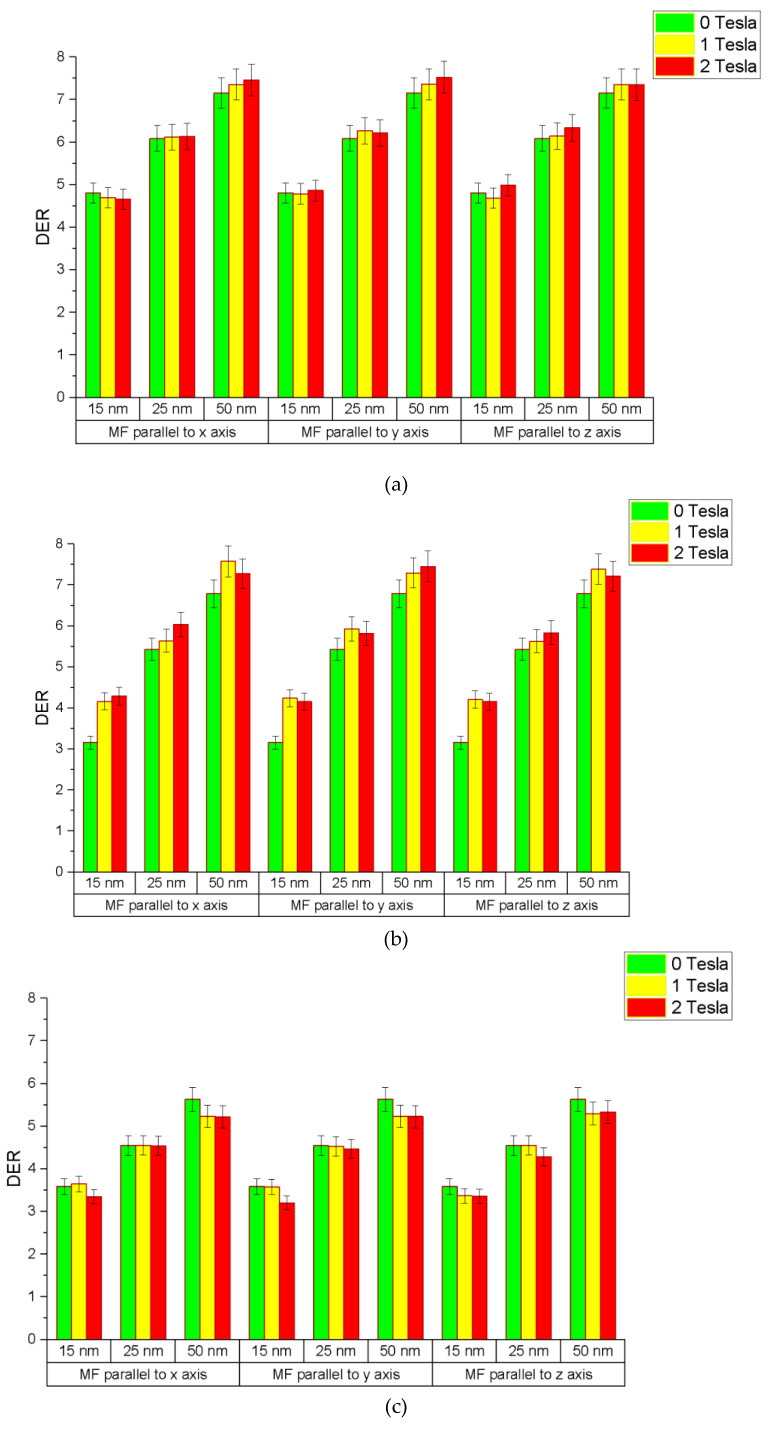
Relationships between the DER and simulation variables of gold nanoparticle size and magnetic field strength and orientation using photon beams with energies equal to (**a**) 50, (**b**) 100 and (**c**) 150 keV.

## Data Availability

Not applicable.

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
