# Peer review of "Gold Nanoparticle DNA Damage by Photon Beam in a Magnetic Field: A Monte Carlo Study"

_nanomaterials, 2021, doi:10.3390/nano11071751_

Round 1

Reviewer 1 Report

General Comments:

This paper presents a Monte Carlo (MC) study of gold nanoparticle DNA damage by photon beams in a magnetic field. The simulated set up is somewhat simplistic in nature and does not reflect a realistic clinical scenario. In particular the paper models irradiation in three kilovoltage photon energies, where dose enhancements factors from nanoparticles have been extensively studied in the literature. However, MRI guided radiotherapy occurs in MV photon irradiation and therefore the validity of the beam energies chosen in this study is questionable. Likewise the choice of magnetic field strength does not correlate with existing commercial MRI-linear accelerator radiotherapy systems. The MC study also considers a single nanoparticle and the effect on DAN damage from this, some justification of this simulated set up should be provided.

Specific Comments

Page 1 Paragraph 1: It should be made clear that that the use of gold nanoparticles is still not routine in clinical practice. Add a reference on this point and a potential roadmap to clinical implementations e.g. Ricketts et al Brit J Radiol 2018, 91, 20180325

Page 1 Paragraph 1: It should also be clarified that  the increased radiosensitivity reported in various studies is the result of a combination of the physical dose enhancement and additional chemical and biological effects associated with the nanoparticle in use. Again add a reference on this point e.g. Her et al Asv Drug Deliv Rev 2017, 109, 84-101

Introduction: Please add some background on how the nanoparticles are administered and the likely uptake of the nanoparticles in vivo to justify your simulated set up.

Simulations: The field strengths of current commercial MRI linear accelerators are 1.5T (Unity Elekta), 0.35T (Viewray), 1T (Australian), 0.5 T (MagnetTx) with respective photon energies 6MV, Co or 6MV, 4 or 6 MV, and 6MV. Please consider the validity of the photon energies and magnetic field strengths simulated in this manuscript and whether the listed values above would be more representative of actual clinical situations.

Page 6 line 156: Please consider and state the uncertainty in your simulations and whether results may be stated to two decimal points given your estimated uncertainty.

Discussion and Conclusion: Further discussion on what the reported work adds to the literature is required, in particular any limitations in this study and how the results add to the current literature and the potential application in clinical practice.

Author Response

Reviewer 1

We would like to thank the Reviewer for the insightful and decent comments to improve this work. Our responses are in the blue font while corresponding corrections/modifications are in red font in the revised manuscript.

General Comments:

This paper presents a Monte Carlo (MC) study of gold nanoparticle DNA damage by photon beams in a magnetic field. The simulated set up is somewhat simplistic in nature and does not reflect a realistic clinical scenario. In particular the paper models irradiation in three kilovoltage photon energies, where dose enhancements factors from nanoparticles have been extensively studied in the literature.

Authors: Thank you very much for your comment. This work is a nanodosimetric study focusing on the DNA damage due to photon interaction with gold nanoparticle under a magnetic field. For simulation related to clinical study, macroscopic approach (mm to cm scale) would be used to determine tumour control. Although dose enhancement have been widely studied, there is no similar study regarding a Monte Carlo model consisting of a nanoparticle and DNA in a magnetic field. Please consider the complexity of the simulation required to track every primary and secondary electrons down to an energy range of eV. This makes the simulation complicated, though the model looks simple.

However, MRI guided radiotherapy occurs in MV photon irradiation and therefore the validity of the beam energies chosen in this study is questionable. Likewise the choice of magnetic field strength does not correlate with existing commercial MRI-linear accelerator radiotherapy systems. The MC study also considers a single nanoparticle and the effect on DAN damage from this, some justification of this simulated set up should be provided.

Authors: When considering the photon beam generated by the linear accelerator to the patient, the energy is MV. However, when the beam entered from the patient’s skin surface and reached the tumour, beam attenuation would happen leading to a reduction of photon energy. Therefore, beam energy of kV in a cellular environment is possible in the simulation. In our study, we used 0-2 T for the magnetic field strength, which is close to the range of the current commercial MRL. Of course, we are not going to mimic any of them from different manufacturers. On the other hand, single-particle geometry is the base of the Monte Carlo simulation model, because the more complicated multi-particle geometry is built from it. Here, we would like to provide results arising from a study on a single particle so that multi-particle model can be constructed in the future.

Specific Comments

Page 1 Paragraph 1: It should be made clear that that the use of gold nanoparticles is still not routine in clinical practice. Add a reference on this point and a potential roadmap to clinical implementations e.g. Ricketts et al Brit J Radiol 2018, 91, 20180325

Authors: Done with the suggested reference added to the revision.

“Many preclinical works have been done and clinical trials are being conducted that building a potential roadmap to clinical implementations [16].”

  1. Ricketts K, Ahmad R, Beaton L, Cousins B, Critchley K, Davies M, Evans S, Fenuyi I, Gavriilidis A, Harmer QJ, Jayne D. Recommendations for clinical translation of nanoparticle-enhanced radiotherapy. Brit J Radiol. 2018 Dec;91(1092):20180325.

Page 1 Paragraph 1: It should also be clarified that the increased radiosensitivity reported in various studies is the result of a combination of the physical dose enhancement and additional chemical and biological effects associated with the nanoparticle in use. Again add a reference on this point e.g. Her et al Asv Drug Deliv Rev 2017, 109, 84-101

Authors: Done with the suggested reference added to the revision.

“This increase of radiosensitivity is due to a combination of the physical dose enhancement and additional chemical and biological effects associated with the nanoparticle [4].”

  1. Her S, Jaffray DA, Allen C. Gold nanoparticles for applications in cancer radiotherapy: Mechanisms and recent advancements. Adv Drug Deliv Rev. 2017 Jan 15;109:84-101.

Introduction: Please add some background on how the nanoparticles are administered and the likely uptake of the nanoparticles in vivo to justify your simulated set up.

Authors: We added statements concerning the nanoparticle transport with reference.

“Gold nanoparticles can transport to the living cells through a liposome-based system. This makes the treatment delivery of gold nanoparticle-enhanced radiotherapy possible [5].”

  1. Chithrani DB, Dunne M, Stewart J, Allen C, Jaffray DA. Cellular uptake and transport of gold nanoparticles incorporated in a liposomal carrier. Nanomedicine: Nanotechnology, Biology and Medicine. 2010 Feb 1;6(1):161-9.

Simulations: The field strengths of current commercial MRI linear accelerators are 1.5T (Unity Elekta), 0.35T (Viewray), 1T (Australian), 0.5 T (MagnetTx) with respective photon energies 6MV, Co or 6MV, 4 or 6 MV, and 6MV. Please consider the validity of the photon energies and magnetic field strengths simulated in this manuscript and whether the listed values above would be more representative of actual clinical situations.

Authors: This work is not investigating clinical MRL which is a macroscopic simulation study. However, we selected 1 and 2 T as the magnetic field strength, because they are close to the range of field strength of various MRL from different manufacturers. For the photon beam energy, we are concerned that the photon energy would interact with the nanoparticle in the cell. These photons can be tracked originally from the primary photon beam generated by the MRL.

Page 6 line 156: Please consider and state the uncertainty in your simulations and whether results may be stated to two decimal points given your estimated uncertainty.

Authors: We agree with the Reviewer and stated the results to one decimal point.

Discussion and Conclusion: Further discussion on what the reported work adds to the literature is required, in particular any limitations in this study and how the results add to the current literature and the potential application in clinical practice.

Authors: We added the following paragraph in the Conclusion section:

“Results in this work provided important information concerning the variation of energy deposition in the DNA, when a magnetic field is present in an irradiated gold nanoparticle. Moreover, this single-nanoparticle model can act as a base for the construction of a more complicated multi-nanoparticle-model focusing on a more realistic cellular environment for clinical practice.”

Reviewer 2 Report

This is an interesting study to see if magnetic field influences the radiological dose enhancement with x-rays and gold nanoparticles. The authors clearly explain the experiment and the results, it is well written and constructed.

It's not exactly clear how many individual photons and electron paths were analyzed to contruct the results. How does this number affect the results? Also, can the authors estimate an error bar in the graphs?

Why did the authors choose for this Au NP - DNA distance? How does it influence the results?

It seems that putting patients in a magnetic field doesn't present great benefits. That in itself is a valuable conclusion. Is this something the authors could have expected, based on the force that the magnetic field exerts on the secondary electrons? Could the authors give some back-of-the-envelope calculations for the amount of magnetic momentum transfer that would be expected?

The authors report small magnetic force enhancements, but in a real world situation, won't these effects cancel each other out, because the nanoparticles are randomly oriented with respect to the DNA?

Author Response

Reviewer 2

We would like to thank the Reviewer for the insightful and decent comments to improve this work. Our responses are in the blue font while corresponding corrections/modifications are in red font in the revised manuscript.

This is an interesting study to see if magnetic field influences the radiological dose enhancement with x-rays and gold nanoparticles. The authors clearly explain the experiment and the results, it is well written and constructed. It's not exactly clear how many individual photons and electron paths were analyzed to contruct the results. How does this number affect the results? Also, can the authors estimate an error bar in the graphs?

Authors: We added the following statements to clarify the number of history and related uncertainty in this study:

“The number of history for the photons interacting with the gold nanoparticle was equal to 300 million in this study, and more photons (~ 2,000 million) were required to achieve similar uncertainty (2%-5% standard deviation) for the simulation model without gold nanoparticle.”

We added error bars in the graphs (Figure 3(a), 3(b) and 3(c)).

Why did the authors choose for this Au NP - DNA distance? How does it influence the results?

Authors: The GNP-DNA distance is based on Ref [35], which found that the DER increased with a decrease of GNP-DNA distance. This is reasonable because the closer the GNP and DNA, the more secondary electrons can reach the DNA to produce damage.

It seems that putting patients in a magnetic field doesn't present great benefits. That in itself is a valuable conclusion. Is this something the authors could have expected, based on the force that the magnetic field exerts on the secondary electrons?

Authors: Although the presence of magnetic field cannot benefit the dosimetry in radiotherapy, the MR imaging can help to visualize the tumour during the dose delivery, treatment planning and patient setup. The presence of variation of electron track affected by the magnetic field is what we expected, and we would like to investigate how such variation would influence the DNA damage.

Could the authors give some back-of-the-envelope calculations for the amount of magnetic momentum transfer that would be expected?

Authors: The magnetic momentum transfer have been considered in the Monte Carlo simulation in this study.

The authors report small magnetic force enhancements, but in a real world situation, won't these effects cancel each other out, because the nanoparticles are randomly oriented with respect to the DNA?

Authors: Thank you so much for such an insightful comment. To investigate this issue, a multi-nanoparticle model is required. It is very complicated to construct such a multi-particle model into the current simulation, but we will add this as our future work (mentioned in the Conclusion section).

“Moreover, this single-nanoparticle model can act as a base for the construction of a more complicated multi-nanoparticle model focusing on a more realistic cellular environment for clinical practice.”

Round 2

Reviewer 1 Report

The authors have addressed or rebutted my comments satisfactorily. The authors have added a sentence in the conclusion explaining that this work could act as a base for a more complex scenario. However, I would suggest the authors also consider adding to the discussion section a little on what would be a more realistic, albeit complex, clinical scenario, and explain the context of their work in relation to this. My initial comments and the rebuttal may form the basis for such an addition to the discussion

Author Response

The authors have addressed or rebutted my comments satisfactorily. The authors have added a sentence in the conclusion explaining that this work could act as a base for a more complex scenario. However, I would suggest the authors also consider adding to the discussion section a little on what would be a more realistic, albeit complex, clinical scenario, and explain the context of their work in relation to this. My initial comments and the rebuttal may form the basis for such an addition to the discussion

Authors: We followed the suggestion to add a paragraph in the Discussion section with a reference concerning macroscopic Monte Carlo simulation:

This work focused on the nanodosimetric change of the DNA interacting with an irradiated gold nanoparticle in the presence of a magnetic field. Based on the results in this study, it is worthwhile to further investigate the dependence of DER on multiple nanoparticles with different distribution patterns, sizes and shapes. Moreover, macroscopic Monte Carlo simulation [39] can be carried out to investigate the dose enhancement of tumour in a patient treated by magnetic resonance guided radiotherapy. This multi-scale study can help us to understand in detail the impact of magnetic field on cancer control from the nanometer to centimeter scale.

  1. Martelli S, Chow JCL Dose enhancement for the flattening-filter-free and flattening-filter photon beams in nanoparticle-enhanced radiotherapy: A Monte Carlo phantom study. Nanomaterials 2020;10:637.